# LEARNING STATE REPRESENTATIONS IN COMPLEX SYSTEMS WITH MULTIMODAL DATA

## ABSTRACT

Representation learning becomes especially important for complex systems with multimodal data sources such as cameras or sensors. Recent advances in reinforcement learning and optimal control make it possible to design control algorithms on these latent representations, but the field still lacks a large-scale standard dataset for unified comparison. In this work, we present a large-scale dataset and evaluation framework for representation learning for the complex task of landing an airplane. We implement and compare several approaches to representation learning on this dataset in terms of the quality of simple supervised learning tasks and disentanglement scores. The resulting representations can be used for further tasks such as anomaly detection, optimal control, model-based reinforcement learning, and other applications.

## 1 INTRODUCTION

In order to act in real world scenarios and control a complex system such as an airplane, car, or industrial facility, an automated agent needs to process complex high-dimensional data coming from different domains: video feeds from cameras, LIDAR sensors on a car, altitude and speed sensors on an airplane, sensors related to the internal state of the system, and so on. An important problem in this regard would be to map this rich stream of multimodal information into a lower-dimensional space that compresses all modalities into a uniform latent representation (embedding); the agent could then use this embedding to learn or otherwise construct control algorithms. Thus, representation learning lies at the heart of optimal control for complex systems with multimodal features.

Over the last decade, deep neural networks have surpassed other methods in processing nearly all modalities of high-dimensional unstructured data, including images, natural language texts, sounds, and time series. One of the most important properties of neural networks that has made the deep learning revolution possible is their ability to extract meaningful low-dimensional representations of raw unstructured input data. Representation learning with deep neural networks is a large and well-established area of research (Kingma & Welling, 2013; Hinton & Salakhutdinov, 2006; Radford et al., 2015). Latent features learned by deep neural networks find applications in various domains, including reinforcement learning for complex systems (Zhang et al., 2018; Watter et al., 2015; van Hoof et al., 2016). In these works, the authors often propose special techniques and architectures to design the latent space in different ways suitable for further use: make the latent space locally linear, capture the dynamics, and so on. Designing such feature extractors is a complex task, usually done by hand. Moreover, it is hard to compare different architectures, to a large extent because it is far from obvious how to measure the quality of resulting representations. One reason for that is that the best metric for the quality of learned representations would be the quality of the final task in question, which is hard to obtain in reinforcement learning due to the sheer scale of this final task.

A common way to deal with such problems is to use a unified dataset and a unified set of benchmarks, such as, for example, ImageNet and the ILSVRC benchmark in computer vision (Russakovsky et al., 2015). Such unified datasets might also prove useful for transfer learning tasks. However, the field of reinforcement learning for complex systems is yet to agree on a common representative large-scale dataset. In this work, we present such a multimodal dataset for the representation learning problem together with a unified benchmark framework for feature extractors suitable for a comprehensive comparative evaluation of different feature extractors. This dataset has been gathered with large-scale computer simulations based on the X-Plane simulator and consists of data streams from various

sensors along with images taken from the frontal camera of the plane. We propose and implement several different metrics for comparison between extracted features. We also survey, implement, and compare different neural architectures for learning multimodal state representations.

The paper is organized as follows. In Section 2 we survey related work on representation learning and evaluation of representations. Section 3 presents the X-Plane dataset and explains its characteristic features. In Section 4, we present the various representation learning models that we have implemented and compared on this dataset. Section 5 shows the evaluation metrics and presents a large-scale comparison across different models, and Section 6 concludes the paper.

## 2 RELATED WORK

Modern agent control methods commonly use techniques based on deep learning as feature extractors to deal with complex multimodal data, either explicitly (Zhang et al., 2018; Watter et al., 2015) or implicitly (Mnih et al., 2015). Explicit techniques separate representations of the learning environment from the agents operating in this enviroment; the environment representation can be learned either independently (Zhang et al., 2018) or in a common end-to-end architecture (Parisi et al., 2017). There are several major approaches to constructing such models (Lesort et al., 2018): (1) *autoencoders* that reconstruct the input observation data, producing the latent representation between encoder and decoder; (2) *forward models* that predict the next state either in the latent space or in the raw data, basing the prediction on the current latent representation; (3) *inverse models* that use two states to predict the actions between them; this approach can be combined with forward models; (4) *models with prior knowledge* of the system that impose additional constraints on the latent space according to some fundamental system properties such as causality, temporal continuity, or some more specific knowledge about the system (Jonschkowski & Brock, 2015); (5) *multimodal temporal fusion* models such as the Memory Fusion Network (Zadeh et al., 2018).

If there is no additional data labeling available such as, e.g., a list of factors, the most direct way to measure the quality of representations would be by evaluating the final quality of solving the main task that a model can achieve based on this representation. However, in real world cases, in particular in reinforcement learning, the main task is often hard to solve and unstable to train, so it cannot be consistently used to evaluate latent representations.

Therefore, many indirect ways have been proposed to measure the quality of representations that introduce proxies that can be expected to lead to better solutions of the final control problem. The most common indirect approaches include (see also a comprehensive survey by Lesort et al. (2018)): (1) *task performance*, the most intuitive metric, where representation quality is measured by the quality of performing some other relatively simple task, e.g., by predicting some available target variables with simple models that take the latent representation as input (Higgins et al., 2016; van Hoof et al., 2016); (2) *KNN-MSE*, proposed by Lesort et al. (2017), measures the degree of preservation of the same neighbors between the latent space and the ground truth: $\text{KNN-MSE}(I) = \frac{1}{k} \sum_{I' \in \text{KNN}(I,k)} \|\phi(I) - \phi(I')\|$, where $I$ is the initial raw input, $I'$ is a neighbour of $I$, and $\phi$ is the feature extractor; KNN-MSE is a good metric in situations where the distance in the original input space is well-defined but becomes hard to apply for highly variable multimodal data; (3) a similar approach with humans in the loop is to evaluate whether similar input states according to human evaluation do indeed map to close representations in the latent space (Sermanet et al., 2018); however, for complex multimodal inputs this is again inapplicable since a human would not have an intuitive notion of similarity between two sets of several hundred sensor readings; (4) *disentanglement scores* (Eastwood & Williams, 2018; Higgins et al., 2016) measure the disentanglement (mutual independence) of extracted features; if there are some known generative factors, these metrics assess whether individual elements of the latent representation capture individual generative factors independently; we will consider such metrics in detail in Section 5.1.

As a representative multimodal dataset for a complex system, we have used the X-Plane flight simulator (Laminar Research, 2018), well-known for its faithful simulation of all systems of an aircraft. It has already been used to solve optimal control problems for aircraft; e.g., Bittar et al. (2014) develop separate control laws for stable and maneuvering flight, and Garcia & Barnes (2009) use X-Plane to simulate a system of several unmanned aerial vehicles. However, to the best of our knowledge this is the first attempt to produce a large-scale dataset for representation learning from X-Plane or any other flight simulator.

## 3 THE X-PLANE DATASET

We present the X-Plane Dataset (Dataset, 2018; Code, 2018) as a benchmark for evaluating the quality of state representations (embeddings), where the main problem is to encode the state of some complex system, learn a mapping from high-dimensional multimodal data to a low-dimensional latent representation. We have used the X-Plane simulation environment because it is extremely accurate and can produce a lot of useful and different sensor data. In total, we have recorded 8011 landings, with mean duration of a landing of 115 seconds. For every landing, the dataset contains the readings of 1090 sensors arranged in time series recorded with with a frequency of 5 frames per second, together with the corresponding image taken from the camera located at the front of the airplane; the images are recorded at the same frequency. Table 1 summarizes the various groups of sensors recorded in the dataset, showing the number of sensors in a group together with a brief description. The total dataset size is 93GB uncompressed; it contains sensor readings and about 7M $256 \times 256$ images, joined into per-flight videos for better compression. To ensure diversity in the dataset, we recorded landings with random perturbations and in different environments: with different airports and runways, time of day, weather conditions etc. We have used at most 12 perturbations (abrupt changes in the environment such as, e.g., wind gusts, malfunction of various systems in the aircraft, and so on) applied during each flight; Table 2 summarizes various failures and perturbations. We have used a custom control unit based on the Boeing 737 guide (Brady, 2014) in order to make automatic landings stable.

During dataset collection, for every landing we chose a random airport and a random runway and spawned a plane in a random position, usually at a distance of 3-5 miles from the runway. There are, in total, 114 different runways in 70 airports used in the dataset. Then we applied different landing conditions—landing speed, flaps position, time of day, visibility, precipitation and so on—and enabled the autopilot that lowered the plane on the glide slope path. To achieve a better landing, we disable the autopilot before touchdown, increase the pitch of the plane, and after touchdown we decrease the pitch and enable reverse thrust. At the end of the flight, there are three possible situations: successful landing, aircraft crash, or time limit reached. We set a strict time limit of 160 seconds for every flight. We have also done an airport-stratified split of the dataset into 4 parts: training set for feature extractors, validation set used for early stopping, training set for benchmark supervised learning models, and test set for scoring the benchmark models. We have made the dataset available at (Dataset, 2018).

## 4 MODELS

For experimental evaluation of different representation learning models, we used a wide variety of different models, from basic autoencoders to dynamic actions-aware encoders. In this section, we present these models, from simple to more complex ones. Our models were constructed with four core building blocks: standard recurrent LSTM cell (Hochreiter & Schmidhuber, 1997), one-dimensional convolutions (Conv1D), ConvLSTM (Shi et al., 2015), and an attention cell (Bahdanau et al., 2014). The attention cell $A(\mathbf{x})$ operates as follows (see Fig. 2):

$$\mathbf{v}_i = \mathrm{ReLU}(\mathbf{W}\mathbf{x}_i + \mathbf{b}), \ \mathbf{s} = \mathrm{softmax}(\mathbf{c}^\top \mathbf{V} + \mathbf{b}), \ \mathbf{a} = \mathbf{V}\mathbf{s}, \ \mathrm{Out} = (\mathbf{x}, \mathbf{a}),$$

where $\mathbf{x}$ is the input, $\mathbf{W}$ is a $d \times N$ matrix for embedding size $d$ and input dimension $N$, $\mathbf{V}$ is the matrix with columns $\mathbf{v}_i$, $\mathbf{c}$ is a vector of size $d$, $\mathbf{s}$ are attention weights. We considered the following specific models (Table 4 lists all models with brief descriptions and numerical results).

**One-dimensional autoencoders.** A simple autoencoder that trains to reconstruct a given set of timesteps (Fig. 2a). We used four types of autoencoders (AE) in the comparison: autoencoder with 1-layer LSTM cells for encoder and decoder, with 2-layer LSTM cells, with a 6-layer convolutional autoencoder with kernel size 7. In LSTM autoencoders, after the encoder block we used simple averaging of vectors from all timestamps in the time series, using the result as embedding. In Conv1D, after the encoder block we used max-pooling to get a single vector from the time series; in the decoder, it is copied the necessary number of times and fed to another Conv1D block. In our experiments, we used 6 layers of Conv1D with kernel size 7.

**Image Autoencoders.** We trained two image autoencoders for reconstructing individual pictures from the flight time series. The first autoencoder uses a PCA encoder and a PCA decoder. To obtain

| Group | Description | # |
|---|---|---|
| EFIS | Data taken from the Electronic Flight Instrument System (EFIS) | 10 |
| annunciators | Signals from annunciator panel: oil/fuel pressure, fuel quantity etc. | 51 |
| autopilot | Autopilot information: autopilot state, heading, airspeed etc. | 46 |
| clock_timer | Various date and time info | 9 |
| controls | Interactions with controls | 11 |
| electrical | State of electrical systems: no. of batteries, bus voltage/load, lights etc. | 141 |
| engine | Engine info: fuel flow, oil quantity, prop speed etc. | 85 |
| fuel | Fuel-related sensors: fuel level, states of tanks etc. | 11 |
| gauges | Gauges info: rate-of-turn, height, roll etc. | 53 |
| gps | GPS course and the index of the navigation aid (NAVAID) | 2 |
| gyros | Data from gyroscopes: indicated pitch, magnetic heading, roll etc. | 52 |
| hydraulics | Hydraulic fluid quantity | 4 |
| ice | De-icer state | 5 |
| pressure | Various pressure-based metrics such as desired attitude and bleeding air | 13 |
| radios | Parameters of interaction with beacons and airports over the radio | 469 |
| switches | Current state of various switches | 48 |
| tcas | Position of other planes | 9 |
| temperature | Outside air temperature | 6 |
| transmissions | Transmission oil pressure and temperature | 2 |
| warnings | Various warnings | 49 |
| other | — | 13 |

Table 1: Groups of sensors represented in the X-Plane dataset.

| DataRef name | Description | # |
|---|---|---|
| rel_engfir0 | Engine 1 is on fire | 1181 |
| rel_engfir1 | Engine 2 is on fire | 1199 |
| rel_gls | Autopilot has lost the Glide Slope | 1006 |
| rel_bird_strike | Bird has hit the plane | 449 |
| rel_rwy_lites | Runway lights inoperative | 1790 |
| frm_ice | Left wing is covered with ice (fraction of icing on wings/airframe) | 553 |
| frm_ice2 | Right wing is covered with ice (fraction of icing on wings/airframe) | 583 |
| rel_servo_ailn | Ailerons servos failed | 976 |
| rel_servo_elev | Elevators servos failed | 1026 |
| rel_servo_thro | Throttles servos failed | 993 |
| rel_engfai0 | Engine 1 has lost power without smoke | 638 |
| rel_engfai1 | Engine 2 has lost power without smoke | 2183 |
| turbulence | Turbulence factor | 3296 |
| wind_speed_kt | Effective wind speed, knots | 1499 |

Table 2: Different kinds of failures (top) and perturbations (bottom) in the X-Plane dataset.

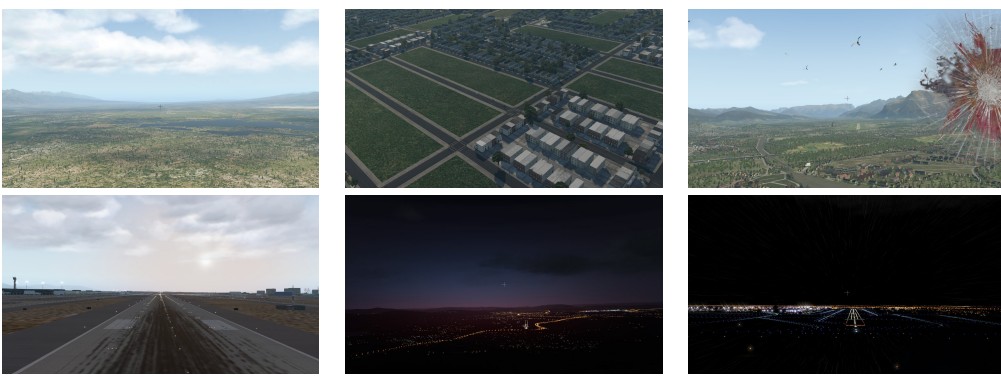

Table 3: Sample images from the dataset.

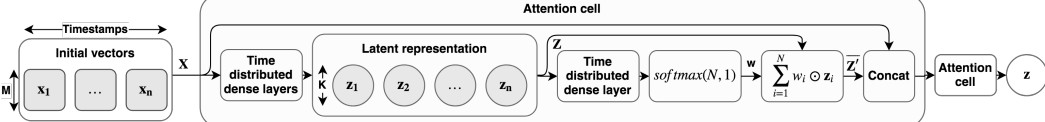

Figure 1: Attention cell architecture.

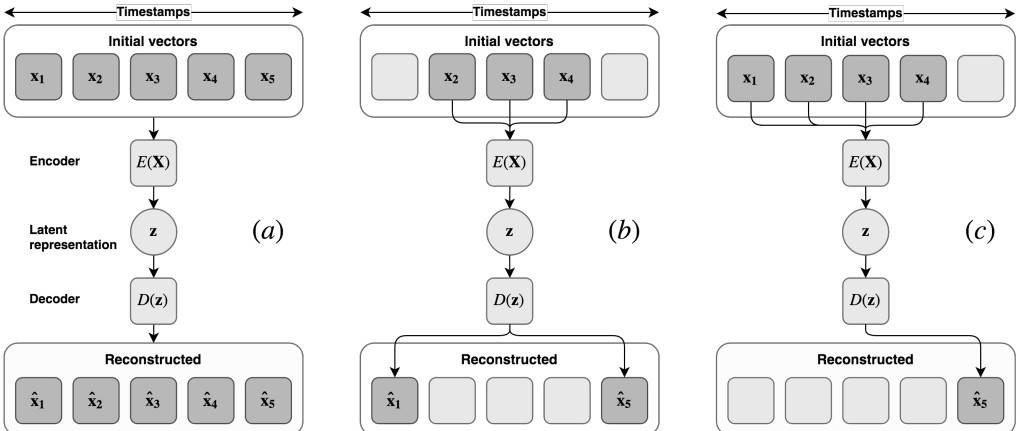

Figure 2: Three main ideas used to construct baseline models: (a) autoencoder for the state vectors, (b) context prediction model, (c) autoregression (forward) model.

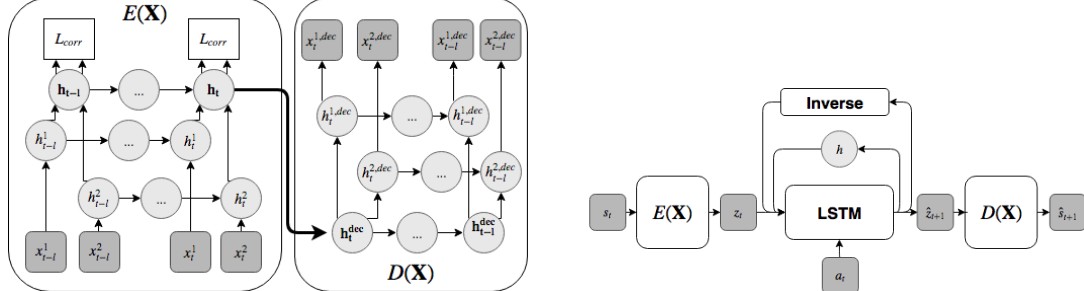

Figure 3: The Multimodal Temporal Encoder.                Figure 4: The Dynamics Model.

the features from a flight sample, we apply the encoder for each image in the sequence and compute the average of all resulting feature vectors. The second architecture is a bit more complicated: its encoder contains the ResNet34 (He et al., 2016) architecture with an additional fully connected layer whose size varies depending on the embedding size $d$, and whose decoder contains fifteen two-dimensional convolutional layers. In contrast to the PCA model, instead of averaging the feature vectors for each flight we used the standard deviation (this class of models worked better empirically in our experiments).

**Autoregression Models.** These models are trained to predict a state at time $t + k$ given some subset of states up to time $t$. In our experiments, we trained autoregression models to predict with $k = 1$ (next time step) and $k = 30$. In this task, as an encoder we have compared LSTM, Conv1D, ConvLSTM, and attention-based architectures, with a simple linear layer after each.

**Context Models.** These models employ ideas similar to the *word2vec* CBOW model (Mikolov et al., 2013). Given a short sequence of timesteps, the model trains to predict several timesteps before and after the sequence; in our experiments, we used window size of 10 before and 10 after given states. To train context models, we have used LSTM, attention cells, and Conv1D blocks. After an encoder block, we repeated an embedding $2C$ times, where $C$ is the context size, and then applied a simple linear layer to each vector with different weights for each of the $C$ timestamps.

**Multimodal Temporal Encoder.** The multimodal temporal encoder (MTE), introduced by Yang et al. (2017), is a state of the art model intended to fuse inputs from different modalities (see Fig. 3). MTE uses an LSTM unit for each modality and shares hidden states across these units. By doing this, MTE forces the model to learn a fused representation across modalities; formally, it adds a correlation loss that computes the correlation between projections of different modalities $L_{\text{corr}}(H_t^1, H_t^2) = \frac{\sum_{i=1}^{M}(h_{ti}^1 - \overline{H_t^1})(h_{ti}^2 - \overline{H_t^2})}{\sqrt{\sum_{i=1}^{M}(h_{ti}^1 - \overline{H_t^1})^2 \sum_{i=1}^{M}(h_{ti}^2 - \overline{H_t^2})^2}}$. The final loss is calculated as

$L = \sum_{i=1}^{M} L_{\text{recon},i} - \lambda L_{\text{corr}}$, where $M$ is the batch size, $h_t^j$ is the representation of modality $j$ at timestep $t$, $H_t^j = \{h_{ti}^j\}_{i=1}^{M}$, $\overline{H}_t^j = \frac{1}{M}\sum_i^M h_{ti}^j$, $\lambda$ is a constant hyperparameter, and $h_t$ is the final embedding (see Fig. 3). The sensors data is encoded with a three-layer fully connected neural network with dropout and ReLU activations, and a pretrained ResNet18 architecture for the images.

**The Dynamics Model.** This approach, proposed by Zhang et al. (2018), introduces a way to decouple the training process (for a reinforcement learning task) into learning the dynamics model and the reward model. The dynamics model (shown in Fig. 4) is trained using a combination of four loss functions: reconstruction loss $L_{t,\text{recon}}(\theta_{\text{enc}}, \theta_{\text{dec}}) = (s_t - \hat{s}_t)^2$, state loss $L_{t,\text{state}}(\theta_{\text{for}}, \theta_{\text{dec}}) = (s_{t+1} - \hat{s}_{t+1})^2$, forward loss $L_{t,\text{for}}(\theta_{\text{for}}, \theta_{\text{enc}}) = (z_{t+1} - \hat{z}_{t+1})^2$, and inverse loss (with a trainable LSTM unit) $L_{t,\text{inv}}(\theta_{\text{inv}}) = (a_t - \hat{a}_t)^2$. The final loss is computed as $L(\theta_{\text{dynamics}}) = \sum_{t=0}^{T}(\lambda_{\text{dec}}(L_{t,\text{recon}} + L_{t,\text{state}}) + \lambda_{\text{for}}L_{t,\text{for}} + \lambda_{\text{inv}}L_{t,\text{inv}})$, where $\hat{z}_{t+1}, h_t = f_{\text{for}}(z_t, a_t, h_{t-1}; \theta_{\text{for}})$, $\hat{a}_t = f_{\text{inv}}(z_t, z_{t+1}; \theta_{\text{inv}})$, $\hat{s}_{t+1} = f_{\text{dec}}(\hat{z}_{t+1}; \theta_{\text{dec}})$, $s_t$ is the state at time $t$, $a_t$ is the action at time $t$, $h_t$ is the hidden state, and $z_t$ is the latent representation at time $t$; $\lambda_{\text{dec}}$, $\lambda_{\text{for}}$, and $\lambda_{\text{inv}}$ are (constant) hyperparameters. If the dynamics change, we need only to re-train the LSTM unit and can keep the encoder and decoder unchanged, with the assumption that already learned representation contains all the necessary information about the new dynamics. Since the original approach was presented only for inputs with a single modality, we expanded this idea for the multimodal case. To encode data from the sensors, we used a three-layer fully connected neural network with dropouts and ReLU activations. For the images, we used a pretrained ResNet18 architecture. After that, a linear layer was used to concatenate outputs of encoders for each modality into the embeddings, and the final embedding results by averaging over the time steps.

## 5 EXPERIMENTAL EVALUATION

### 5.1 EVALUATION FRAMEWORK

As part of the dataset package, we have implemented and made available the framework which is designed to make a comprehensive evaluation of learned representation based on a number of fixed predefined tests. The main idea behind our framework is to combine the two main approaches to measuring representation quality. First, we measure quality of representations by using them as features for a number of simple tasks. This approach was used, in particular, in the Black Box Learning Challenge (Goodfellow et al., 2013) and Unsupervised and Transfer Learning Challenge (Guyon et al., 2011), whose main objective was to learn a good representation from rich unlabeled data, and representations were evaluated on supervised learning tasks that were not known to the participants.

The second approach is to evaluate the *disentanglement* in representations. For this we use a QEDR (Quantitative Evaluation of Disentangled Representations) framework proposed by Eastwood & Williams (2018). The idea is that the ideal representation of data is a vector of separated and independent generative factors for the data (perhaps scaled and permuted). The matrix $R$, where $R_{ij}$ is the relative importance of code (representation) variable $c_i$ in predicting generative factor $z_j$, is used to compute three evaluation metrics for the quality of a representation. *Disentanglement* is a measure of the degree to which a representation factorizes the factors of variation in original data; for a code variable $c_i$ it is defined as $D_i = 1 - H(P_{i.})$ where $H(P_{i.})$ is the entropy of the pseudo-distribution $P_{i.}$; $P_{ij} = R_{ij}/\sum_k R_{ik}$ denotes the "probability" of $c_i$ being important for predicting generative factor $z_j$, and total disentanglement is computed as the weighted average $\sum_i \rho_i D_i$, where $\rho_i = \sum_j R_{ij}/\sum_{kj} R_{kj}$ is the relative code variable importance. *Completeness* is a measure of the degree to which a factor of variation in the original data is captured by a single code variable; for a factor $z_j$ it is defined as $C_j = 1 - H(\tilde{P}_{.j})$ where $H(\tilde{P}_{.j})$ is the entropy of the pseudo-distribution $\tilde{P}_{.j}$; total completeness is the average of $C_j$. *Informativeness* measures the amount of information that a representation captures about the underlying factors of variation; following Eastwood & Williams (2018), we use normalised root-mean-square ($NRMSE$) as the informativeness metric. We use failure scores as generative factors since we know the ground truth for them and they are mutually independent. We define $R_{ij} = |W_{ij}|$, where $W$ is the weight matrix of lasso regression learned on the representation vector to predict the vector of factors.

Our evaluation framework is shown in Fig. 5. We assume that a feature extractor constructs a single vector representation for a time series of sensor readings. We have implemented disentanglement,

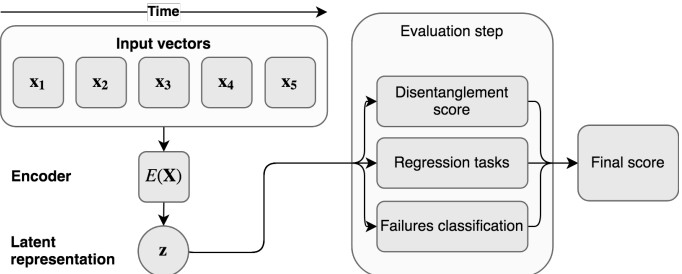

Figure 5: Evaluation pipeline.

completeness, and informativeness scores computed on 12 failures scores as generative factors and 5 supervised learning targets, including: wind speed, turbulence power, vertical acceleration at the time of landing (a quality of touchdown), autoregression (given $t$ states, predict state $t + 1$), failure classification (in dataset generation we randomly applied various failures to the airplane, and the task is to find out whether a failure is present in a given time series). To evaluate supervised learning, we first apply the provided feature extractors for every training example, then train simple models (namely, a three-layer fully connected neural network with ReLU activations) on these features for each target, and finally compute the accuracy on the hold-out set; these accuracies are the final performance metrics. We have made the code for the evaluation pipeline available at (Code, 2018).

## 5.2 EVALUATION RESULTS

Table 4 shows the main results of our evaluation study; in this section we interpret these quantitative results. The models were trained on the X-Plane dataset with regression tasks evaluated on a hold-out set; time series lengths varied from 25 to 75 during training. To establish a simple baseline for our models we use the mean target value over the training set as a prediction. In Table 4, the "MEAN baseline" row shows the absolute MSE values for MEAN and the other rows are normalized to the MEAN baseline. First, we see that different regression tasks have very different models doing well on them; e.g., most models cannot outperform even the mean baseline for wind regression, and models that show good results on wind regression perform poorly on auto regression and vice versa. The effect of images on the results is contradictory: we have trained TS Regression and Dynamics models on three types of data (sensors, images, and both) each, and for TS Regression adding images improves scores on benchmarks, while for the Dynamics model the results deteriorate. The results show that embedding size $d$ needs to be tuned for each model separately; some models even "explode" and give unstable results (very large errors) for some values of $d$; there is no general correlation between $d$ and benchmark scores. It appears that the Dynamics model trained on sensors only is the best tradeoff between efficiency and quality since it is by far the fastest model. Both disentanglement and completeness scores are small, perhaps because we used only a small subset of generative factors to compute them. The disentanglement score is better for smaller $d$, which shows that information is packed more efficiently in this case. Performance of the models that use images strongly depends on the quality of the images themselves. For example, if the horizon and runway are clearly visible (as in Table 5a), models with images outperform models that do not use them (Table 5), while on night-time noisy images (Table 5b) models that use images lose.

## 6 CONCLUSION

In this work, we have presented the dataset for learning state representation and a unified evaluation framework to measure the quality of multimodal joint representations produced by different encoders, through both secondary supervised learning tasks and quantitative metrics of disentanglement and informativeness quality. This work represents an attempt to help advance the research in model-based reinforcement learning and state representation learning by providing a unified. At the same time, the large-scale comparison between baseline and state of the art models that we perform in this work has produced some interesting results by itself. In general, we believe that this dataset can become the new standard for evaluation in representation learning for complex systems, and we hope it will inspire many novel techniques to advance the state of the art that we have attempted to establish and quantify in our practical evaluation.

| Model | Img | $d$ | Wind | Turb. | Land | Auto | Fail | Overall | Inform. | Disent. | Compl. |
|---|---|---|---|---|---|---|---|---|---|---|---|
| MEAN Baseline | | — | 0.969 | 1.036 | 2.936 | 2.039 | 0.961 | 7.940 | — | — | — |
| LSTM Autoencoder | | 32 | 1.014 | 0.985 | 0.976 | 0.953 | 0.788 | 0.953 | 0.912 | 0.095 | 0.107 |
| 1 layer encoder, | | 64 | 1.017 | 0.970 | 0.986 | 0.795 | 0.763 | 0.912 | 0.893 | 0.099 | 0.126 |
| 1 layer decoder | | 128 | 1.096 | 0.954 | 0.934 | 0.786 | 0.724 | **0.893** | **0.885** | 0.067 | 0.121 |
| | | 256 | 1.076 | 1.012 | 8.770 | **0.777** | 0.941 | 3.820 | 0.897 | 0.054 | 0.112 |
| LSTM Autoencoder | | 32 | 1.013 | 0.956 | 0.925 | 0.933 | 0.949 | 0.945 | 0.892 | 0.135 | 0.153 |
| 2 layers bidir. encoder | | 64 | 1.024 | 0.967 | 1.075 | 0.807 | 0.783 | 0.951 | 0.892 | 0.089 | 0.131 |
| 2 layers bidir. decoder | | 128 | 1.062 | 0.978 | 0.958 | 0.832 | 0.797 | 0.921 | 0.889 | 0.070 | 0.124 |
| | | 256 | 1.521 | 4.236 | 0.958 | 0.846 | 1.797 | 1.527 | 0.941 | 0.059 | 0.122 |
| Conv1D AutoEncoder | | 32 | 1.008 | 0.994 | 1.068 | 0.893 | 0.992 | 0.997 | 0.924 | 0.086 | 0.111 |
| 6-layer 1D convolutional | | 64 | 1.002 | 0.982 | 0.960 | 0.862 | 0.894 | 0.935 | 0.924 | 0.078 | 0.126 |
| autoencoder with | | 128 | 1.002 | 0.975 | 0.975 | 0.867 | 0.901 | 0.943 | 0.912 | 0.070 | 0.128 |
| kernel size $7 \times 7$ | | 256 | 1.000 | 0.978 | 1.009 | 0.893 | 0.955 | 0.968 | 0.922 | 0.059 | 0.123 |
| PCA Autoencoder | ✓ | 32 | 0.999 | 0.961 | 0.901 | 0.862 | 0.861 | 0.906 | 0.921 | 0.084 | 0.111 |
| 1 layer decoder | | 64 | 1.000 | 0.997 | 0.997 | 0.997 | 0.999 | 0.999 | 0.917 | 0.075 | 0.119 |
| $256 \times 256 \times 3$ images | | 128 | 1.000 | 1.000 | 0.999 | 0.997 | 0.998 | 0.999 | 0.917 | 0.064 | 0.119 |
| mean features over time series | | 256 | 1.000 | 0.998 | 1.000 | 0.999 | 0.999 | 0.999 | 1.000 | 0.067 | 0.132 |
| ResNet34 Autoencoder | ✓ | 32 | **0.988** | 0.960 | **0.853** | 0.927 | 0.942 | 0.913 | 0.924 | **0.137** | **0.157** |
| ResNet34 encoder with 1 FC layer | | 64 | 1.006 | 0.958 | 0.898 | 0.913 | 0.916 | 0.925 | 0.931 | 0.080 | 0.136 |
| 15 layer conv2d decoder | | 128 | 0.994 | 0.972 | 0.910 | 0.908 | 0.927 | 0.930 | 0.940 | 0.068 | 0.127 |
| images only | | 256 | 0.992 | 0.963 | 0.901 | 0.916 | 0.935 | 0.928 | 0.978 | 0.071 | 0.130 |
| TS Regression LSTM | | 32 | 1.010 | 0.950 | 1.037 | 0.816 | 0.948 | 0.955 | 0.925 | 0.081 | 0.108 |
| 2-layer bidir. LSTM | | 64 | 1.002 | 1.211 | 1.122 | 1.436 | 5.566 | 1.738 | 0.920 | 0.066 | 0.106 |
| trained to predict | | 128 | 1.035 | 1.119 | 0.872 | 0.886 | 1.558 | 1.011 | 0.921 | 0.062 | 0.116 |
| state at $t + 30$ | | 256 | 1.074 | 1.084 | 0.952 | 0.888 | 1.034 | 0.978 | 0.925 | 0.058 | 0.109 |
| TS Regression Attention | | 32 | 1.020 | 0.985 | 0.999 | 0.940 | 0.948 | 0.978 | 0.920 | 0.084 | 0.115 |
| 3-layer attention encoder | | 64 | 1.004 | 1.036 | 1.039 | 1.464 | 1.005 | 1.139 | 0.923 | 0.072 | 0.115 |
| trained to predict | | 128 | 1.136 | 0.962 | 0.869 | 1.241 | 0.788 | 0.999 | 0.922 | 0.059 | 0.108 |
| state at $t + 1$ | | 256 | 1.039 | 1.174 | 0.991 | 1.363 | 1.887 | 1.225 | 0.923 | 0.054 | 0.114 |
| TS Regr. ConvLSTM+LSTM | ✓ | 32 | 1.028 | 0.969 | 0.961 | 0.846 | 0.870 | 0.930 | 0.922 | 0.083 | 0.115 |
| 5-layer conv. LSTM with $3 \times 3$ kernel | | 64 | 1.034 | 1.004 | 0.949 | 0.813 | 0.929 | 0.929 | 0.923 | 0.077 | 0.116 |
| for images and 1-layer LSTM for sensors, | | 128 | 1.007 | 1.009 | 1.009 | 1.043 | 1.070 | 1.025 | 0.922 | 0.064 | 0.117 |
| trained to predict state at $t + 30$ | | 256 | 1.018 | 0.977 | 0.970 | 0.791 | 0.857 | 0.917 | 0.915 | 0.055 | 0.114 |
| TS Regr. ConvLSTM | ✓ | 32 | 1.000 | 0.998 | 0.999 | 0.996 | 0.999 | 0.998 | 0.917 | 0.084 | 0.109 |
| 5-layer conv. LSTM with | | 64 | 1.000 | 0.999 | 0.999 | 0.996 | 0.999 | 0.998 | 0.921 | 0.072 | 0.113 |
| $3 \times 3$ kernel for images, | | 128 | 1.000 | 0.998 | 1.000 | 0.999 | 0.998 | 0.999 | 0.921 | 0.062 | 0.112 |
| trained to predict state at $t + 30$ | | 256 | 1.000 | 0.998 | 1.000 | 1.000 | 0.999 | 1.000 | 0.923 | 0.059 | 0.120 |
| Context LSTM Regressor | | 32 | 1.011 | 1.079 | 1.206 | 0.931 | 1.005 | 1.071 | 0.923 | 0.089 | 0.122 |
| 2-layer bidir. LSTM | | 64 | 1.001 | 0.993 | 1.023 | 0.834 | 0.855 | 0.948 | 0.924 | 0.073 | 0.116 |
| trained to predict states | | 128 | 1.271 | 1.059 | 1.262 | 1.045 | 0.906 | 1.138 | 0.922 | 0.059 | 0.110 |
| $[t - 11, t - 1]$ and $[t + 1, t + 11]$ | | 256 | 1.114 | 1.259 | 1.325 | 1.067 | 0.876 | 1.170 | 0.916 | 0.052 | 0.110 |
| Context Attention Regressor | | 32 | 1.148 | 0.950 | 0.931 | 1.103 | 0.820 | 0.991 | 0.924 | 0.079 | 0.107 |
| 3-layer attention net, | | 64 | 1.041 | **0.922** | 0.868 | 1.012 | **0.691** | 0.912 | 0.921 | 0.069 | 0.109 |
| trained to predict states | | 128 | 1.021 | 1.026 | 1.243 | 1.153 | 0.762 | 1.106 | 0.922 | 0.059 | 0.110 |
| $[t - 11, t - 1]$ and $[t + 1, t + 11]$ | | 256 | 1.008 | 0.961 | 0.919 | 1.542 | 0.751 | 1.086 | 0.924 | 0.053 | 0.112 |
| Context Conv1D Regressor | | 32 | 1.001 | 0.979 | 0.977 | 0.858 | 0.893 | 0.940 | 0.921 | 0.087 | 0.121 |
| 6-layer 1D convolutional | | 64 | 1.002 | 0.978 | 1.000 | 0.842 | 0.863 | 0.940 | 0.923 | 0.071 | 0.116 |
| network trained to predict states | | 128 | 1.002 | 0.970 | 0.978 | 0.829 | 0.883 | 0.930 | 0.923 | 0.062 | 0.115 |
| $[t - 11, t - 1]$ and $[t + 1, t + 11]$ | | 256 | 1.002 | 0.974 | 0.979 | 0.836 | 0.890 | 0.934 | 0.923 | 0.059 | 0.122 |
| Multimodal Temporal Encoder | ✓ | 32 | 1.002 | 0.976 | 0.999 | 0.814 | 0.913 | 0.938 | 0.923 | 0.088 | 0.115 |
| LSTM with shared weights | | 64 | 1.001 | 0.962 | 0.999 | 0.804 | 0.884 | 0.931 | 0.922 | 0.067 | 0.105 |
| between inputs, | | 128 | 1.010 | 0.982 | 1.000 | 0.816 | 0.903 | 0.940 | 0.920 | 0.060 | 0.108 |
| ResNet18 | | 256 | 1.003 | 0.988 | 0.998 | 0.847 | 0.949 | 0.953 | 0.920 | 0.055 | 0.114 |
| The Dynamics Model (X) | | 32 | 1.000 | 0.998 | 0.999 | 0.950 | 0.981 | 0.984 | 0.922 | 0.095 | 0.124 |
| Decouple dynamics with ResNet18, | | 64 | 1.000 | 0.984 | 1.000 | 0.951 | 0.968 | 0.981 | 0.910 | 0.073 | 0.115 |
| embedding is the mean of | | 128 | 1.001 | 0.971 | 0.925 | 0.890 | 0.836 | 0.920 | 0.923 | 0.063 | 0.114 |
| embs. over time series | | 256 | 1.000 | 1.000 | 1.000 | 1.000 | 0.998 | 1.000 | 0.923 | 0.061 | 0.127 |
| The Dynamics Model (Img) | ✓ | 32 | 1.000 | 0.998 | 0.999 | 0.929 | 0.998 | 0.981 | 0.924 | 0.097 | 0.126 |
| Decouple dynamics with ResNet18, | | 64 | 1.000 | 0.998 | 0.999 | 0.996 | 0.998 | 0.998 | 0.923 | 0.074 | 0.119 |
| embedding is the mean of | | 128 | 1.000 | 0.999 | 1.000 | 0.986 | 0.999 | 0.996 | 0.924 | 0.068 | 0.123 |
| embs. over time series | | 256 | 1.000 | 0.998 | 0.949 | 0.897 | 0.881 | 0.940 | 0.925 | 0.060 | 0.124 |
| The Dynamics Model (Img+X) | ✓ | 32 | 1.000 | 0.998 | 0.998 | 0.996 | 0.998 | 0.998 | 0.926 | 0.082 | 0.108 |
| Decouple dynamics with ResNet18, | | 64 | 1.000 | 0.998 | 1.000 | 0.998 | 0.999 | 0.999 | 0.923 | 0.075 | 0.122 |
| embedding is the mean of | | 128 | 1.000 | 0.998 | 1.000 | 0.929 | 0.998 | 0.981 | 0.921 | 0.067 | 0.126 |
| embs. over time series | | 256 | 1.000 | 0.999 | 1.000 | 0.943 | 0.998 | 0.985 | 0.912 | 0.060 | 0.126 |

Table 4: Evaluation results; left to right: model name and description, whether it uses images, latent dimension $d$, mean inference time $t$ (s), MSE for regression tasks normalized to the MEAN baseline (lower is better), QEDR scores (Section 5.1).

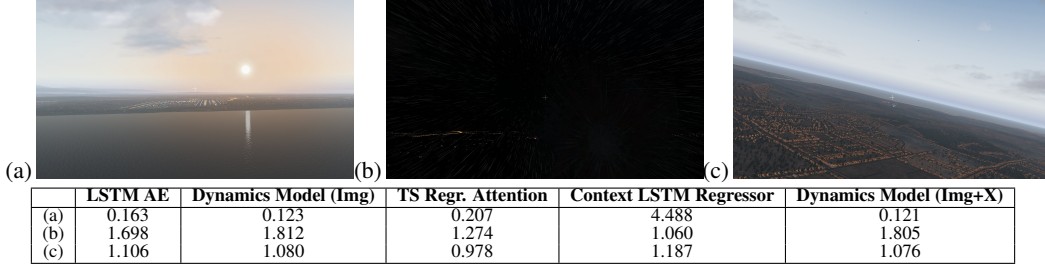

| | LSTM AE | Dynamics Model (Img) | TS Regr. Attention | Context LSTM Regressor | Dynamics Model (Img+X) |
|---|---|---|---|---|---|
| (a) | 0.163 | 0.123 | 0.207 | 4.488 | 0.121 |
| (b) | 1.698 | 1.812 | 1.274 | 1.060 | 1.805 |
| (c) | 1.106 | 1.080 | 0.978 | 1.187 | 1.076 |

Table 5: Images for qualitative evaluation of the models and regression errors; $d = 128$ in all models.

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
