# OpenReview forum: "Learning State Representations in Complex Systems with Multimodal Data"
_ICLR.cc/2019/Conference_

### Official Review · AnonReviewer2 · 2018-11-02
**Potentially impactful work, but lack clarity**

**Rating:** 5
**Confidence:** 4

**Review:**

The work releases a large-scale multimodal dataset recorded from the X-Plane simulation, as a benchmark dataset to compare various representation learning algorithms for reinforcement learning. The authors also proposed an evaluation framework based on some simple supervised learning tasks and disentanglement scores. The authors then implemented and compared several representation learning algorithms using this dataset and evaluation framework.

pros:
1.  Releasing this dataset as a benchmark for comparing representation learning algorithms can potentially impact the community greatly;
2. The authors combined several existing work on measuring representation learning algorithms and proposed an evaluation framework to evaluate the quality of learned representation using supervised learning tasks and disentanglement scores;
3. The authors implemented an extended list of representation learning algorithms and compared them on the dataset;

cons:
1. the paper lacks clarification and guideline to convince the readers of the usefulness of the dataset and the evaluation framework. The authors spent almost half of the space explaining different existing representation learning algorithms. A more convincing story would be to find a few well-established representation learning algorithms to corroborate on the reliability of the dataset and the evaluation metrics;
2. More details should be put into describing the dataset. It is not clear why this dataset is particularly suited for evaluating representation learning in the context of reinforcement learning. Do the authors have insight on the difficulty of the task? While having multi-modality is appreciated, it might worth thinking a separate dataset focusing on a single modality, e.g., image;
3.  Given that the authors designed the dataset for evaluating representation learning for reinforcement learning, it is worth evaluating these algorithms on solving the main task using some standard RL techniques on top of the learned representations.
4. Table 4 is difficult to parse.

---

> ### Author Response · Authors · 2018-11-26
> **Response to the Reviewer 2**
>
>
> > The paper lacks clarification and guideline to convince the readers of the usefulness of the dataset and the evaluation
> > framework. The authors spent almost half of the space explaining different existing representation learning
> > algorithms. A more convincing story would be to find a few well-established representation learning algorithms to
> > corroborate on the reliability of the dataset and the evaluation metrics;
>
> The main contribution of our work is a large-scale multimodal dataset that can be used for representation learning together with a set of baselines and state of the art models evaluated in a common way, with the dataset, models, and evaluation metrics all made available. The representation learning algorithms used in our comparison are indeed well-established, and while we compare across a wide range of architectures, most of them stem from the basic tasks (autoencoder, autoregression) that are commonly used to learn latent representations.
>
> > More details should be put into describing the dataset. It is not clear why this dataset is particularly suited for
> > evaluating representation learning in the context of reinforcement learning. Do the authors have insight on the
> > difficulty of the task? While having multi-modality is appreciated, it might worth thinking a separate dataset focusing
> > on a single modality, e.g., image;
>
> In model-based reinforcement learning, one needs to design an environment representation and a state transition model. While we do not directly address this problem, we argue that our dataset may be useful for such model design iteration, providing means to ensure that the model can perform basic tasks such as reconstruction from latent space or next state/action prediction from the latent representation. Our primary contribution here is that we are the first to deliver a large-scale complex truly multimodal dataset suitable for such tasks, with all modalities aligned directly in the dataset.
>
> > Given that the authors designed the dataset for evaluating representation learning for reinforcement learning, it is
> > worth evaluating these algorithms on solving the main task using some standard RL techniques on top of the learned
> > representations.
>
> This is definitely an important direction for further work. For the paper presenting the dataset and a set of state of the art baselines for further research, we decided to leave the basic underlying tasks and metrics as simple as possible. But yes, we expect the X-Plane dataset to serve as the basis for exactly this kind of work in the future.
>
> > Table 4 is difficult to parse.
>
> Unfortunately, due to space constraints it is hard to provide significantly more explanations; however, we have re-normalized Table 4 to the MEAN baseline to make the numbers more comparable.

---

### Official Review · AnonReviewer1 · 2018-11-03
**main contribution: contribution of multi-modal dataset, evaluation code for learning representation tasks and results on dataset**

**Rating:** 6
**Confidence:** 3

**Review:**

The paper looks into contribution of data set for multi-modal learning using X-Flight simulator in various settings. The authors also contribute code for evaluation of the learning representation tasks and present the results for the data using various setups from autoencoders to dynamics model, using sensor only data and combining image and sensor data, and predicting various timesteps.

Improvements
Multimodal datasets have been made available previously in Image, video, text combinations, where the outcome was clear (for e.g learning caption etc.), however, in this dataset, the task is more challenging (for e.g predicting the various sensor readings or landing outcome). The paper would benefit from
- adding clarification on the Learning tasks, as some of the descriptions/settings and result discussion need more explanation. An e.g predicting the timesteps ahead can be meaning different things, depending on when the start time is, sampling rate and the time to land.
- measure of the scale where only MSE is mentioned for the tasks in the results
- why the time with lower latent dimensions was same as with higher
- the explanation for some of the measure being out of whack for some settings is attributed to challenges with the data set and e.g. is provided for images with nighttime landing. A quantitative number around such cases/for the e.g. in the training data, and test data would be good

---

> ### Author Response · Authors · 2018-11-26
> **Response to Reviewer 1**
>
>
> > adding clarification on the Learning tasks, as some of the descriptions/settings and result discussion need more
> >explanation. An e.g predicting the timesteps ahead can be meaning different things, depending on when the start
> >time is, sampling rate and the time to land.
>
> Section 3 contains a description for the sampling rate and other parameters of the dataset, e.g.: “For every landing, the dataset contains the readings of 1090 sensors arranged in time series recorded with with a frequency of 5 frames per second”. For autoregression problems, we do not track specifically whether, e.g., the plane has already landed in a specific autoregression task and leave it to the model to infer that the plane is going to touch the ground very soon. For problems where we are trying to predict the moment of landing or moment of failures, we have cut off the window in such a way that the landing/failure occurs not at the very end or beginning of the window. We have also added negative examples to the dataset, cutting off windows with no events.
>
> > measure of the scale where only MSE is mentioned for the tasks in the results
>
> Indeed, the MSE scores are not well normalized and may differ in scale between different tasks. To fix this, we have normalized the values in Table 4 to the MEAN baseline.
>
> > why the time with lower latent dimensions was same as with higher
>
> Thank you very much for catching this! Unfortunately, this was due to a defect in our experimental setup: we have used different batch sizes for different models and, alas, even ran some models on two GPUs to speed up training. Therefore, while the inference time results still, in our opinion, can be useful to compare different models with each other, we have to admit they are rather confusing now. Since nothing in our evaluation really hinges on the time, we have removed the “Inference time” column from Table 4.
>
> > the explanation for some of the measure being out of whack for some settings is attributed to challenges with the
> > data set and e.g. is provided for images with nighttime landing. A quantitative number around such cases/for the e.g.
> > in the training data, and test data would be good
>
> Unfortunately, we are not sure we have understood this question in full. Table 2 shows the descriptions for different kinds of failures together with how many times a given failure appears in the dataset. In terms of how a given failure affects the sensors… well, often in a highly complex way which is exactly what we want our representations to be able to capture. For the failures we use, there is no single sensor in the dataset that would just flip to 1 when a given failure occurs.

---

### Official Review · AnonReviewer3 · 2018-11-03
**a new dataset and evaluation framework for learning representations for landing an airplane**

**Rating:** 6
**Confidence:** 3

**Review:**

Overview and contributions:
The authors present a newly collected dataset and evaluation framework for learning representations for landing an airplane. The dataset is collected from the X-Plane simulation environment and consists of 8011 landings, each landing consists of time series data from 1090 sensors. Their evaluation metric is a combination of disentanglement score, regression tasks, and failure classification. The authors test a combination of baseline models from basic autoencoders to dynamic actions-aware encoders. The writing is generally clear but I have doubts about the proposed evaluation metrics, experiments, and significance of the dataset (details below).

Strengths:
1. The task seems to be novel and complex. The authors have done a good job of collecting the dataset and ensuring that the data is clean and comprehensive.
2. The authors have performed a comprehensive job of evaluating the combinations of baseline models for their proposed task.

Weaknesses:
1. Table 4 on evaluation results, while comprehensive, lacks some explanation. The issue with MSE is that it is hard to interpret what these values mean. Specifically, how difficult is this task? How well can a human perform on this task? How well are the baselines doing relative to human-level performance, and is there room for improvement? The answers to these questions are important towards whether this new dataset will be a strong benchmark for representation learning.
2. There is less novelty in terms of the models presented for evaluation since they are composed of existing models. What are some state-of-the-art models for similar tasks, and do they constitute fair comparison?

Questions to authors:
1. Refer to weakness points 1 and 2.
2. What biases do you think might exist in the dataset during the collection process? How might these biases affect what the models learn, and how can they be mitigated?
3. How do you ensure that all sensors are active at all times and that all sensors provide useful information for predicting the label? Are there cases where the multisensor data is noisy in certain modalities or missing in other modalities? If so, what are some models that can remain robustness to noisy or missing modalities?
4. Why do you think disentangled representations will help? Sure, they have been generally shown to help learn more interpretable representations, and help in flexible generation from disentangled factors. But in terms of discriminative or generative performance on your newly proposed dataset, does learning disentangled representations help? What are some models that can learn effectively learn such disentangled representations?

Presentation improvements, typos, edits, style, missing references:
Section 3, line 7, 'with with a frequency' -> 'with'
I would suggest referring to some recent work on multimodal temporal fusion, such as "Memory Fusion Network for Multi-view Sequential Learning. Amir Zadeh, Paul Pu Liang, Navonil Mazumder, Soujanya Poria, Erik Cambria, Louis-Philippe Morency, AAAI 2018"

---

> ### Author Response · Authors · 2018-11-26
> **Response to Reviewer 3**
>
>
> > Table 4 on evaluation results, while comprehensive, lacks some explanation. The issue with MSE is that it is hard to
> > interpret what these values mean. Specifically, how difficult is this task? How well can a human perform on this
> > task?
>
> We include the MEAN baseline for that purpose. It is difficult to make a fair comparison with human performance on these tasks since normally a human pilot will have access to full information on some factors from the cockpit (such as wind) or can reason about other factors from experience (unavailable in the training set) and better predict the state at time moment t+k. In this sense, our benchmarks measure a model's ability to catch the hidden factors from the observations.
>
> In the paper, we have added a clarification on the MEAN baseline:
> “To establish a simple baseline for our models we use the mean target value over the training set as a prediction. In Table 4, the ``MEAN baseline'' row shows the absolute MSE values for MEAN and the other rows are normalized to the MEAN baseline.”
>
> > There is less novelty in terms of the models presented for evaluation since they are composed of existing models.
> > What are some state-of-the-art models for similar tasks, and do they constitute fair comparison?
>
> There are not too many models currently available for multimodal temporal data. We have implemented and compared two state of the art models:
> -- Decouple Dynamics (Zhang et al., 2018; arXiv:1804.10689) proposes the method to explicitly extract embeddings from temporal data and then train a RL agent on such representations;
> -- Multimodal Temporal Encoder (Yang et al., 2017; arXiv:1704.03152) proposes an encoder specifically designed for multimodal temporal data (originally for video and audio streams).
>
> However, as the evaluation table (Table 4) shows, these models are not necessarily the best for any given task..
>
> > What biases do you think might exist in the dataset during the collection process? How might these biases affect
> > what the models learn, and how can they be mitigated?
>
> We use the same aircraft and the same valid landing settings, which in particular means that the initial weight of the aircraft is the same and changes only as the fuel gets burned. This can be used to compute the flight altitude and distance to start point. A change in the initial weight might influence the representation quality, and to avoid it we varied the starting location of the aircraft (so that the weight would be different by the time landing begins). Also, for an incorrect landing (crash) into the forest or the city, the airplane does not stop but keeps moving through objects until simulation is over; this is a feature of X-Plane, but in our experiments it has not made a difference for learning representations.
>
> Generally, we have tried to avoid potential biases by adding a large number of airports and runways.
>
>
> > How do you ensure that all sensors are active at all times and that all sensors provide useful information for
> > predicting the label? Are there cases where the multisensor data is noisy in certain modalities or missing in other
> > modalities? If so, what are some models that can remain robustness to noisy or missing modalities?
>
> We have selected the most highly variable sensors, excluding the dead ones (the ones that do not change during the flight). We have also ensured that the factors we predict in benchmarks are related to sensor readings. Actually, all benchmarking factors affect plane dynamics that can be (theoretically speaking) estimated from the provided time series. We do not model sensor faults, modeling of sensor noise is left to the X-Plane simulator, and normally the sensors work without any interruptions in the simulation.
>
> In future work, we plan to investigate the stability of different models to missing modalities or suddently increased noise in some sensors.
>
>
> > Why do you think disentangled representations will help? Sure, they have been generally shown to help learn more
> > interpretable representations, and help in flexible generation from disentangled factors.
>
> In machine learning, disentanglement of different factors in a representation is generally useful when (1) there are many different tasks the latent representation might serve for, (2) some of these tasks are not known in advance, and (3) there is a need to understand and/or present to the user the underlying factors behind a decision; see, e.g., (Bengio et al., 2014, arXiv:1206.5538; Chen et al., 2016, arXiv:1606.03657). All three factors are present in our case: we present a wide set of tasks in the paper already, and the ultimate purpose would be to use the resulting representations for control purposes, i.e., for tasks that are not well defined in advance at all.
>
>
> > Presentation improvements, typos, edits, style, missing references:
>
> Fixed and added, thank you!

---

> > ### Comment · AnonReviewer3 · 2018-12-06
> > **good job**
> >
> > The rebuttal addressed my main concerns and I am happy to increase my rating from 5 to 6.
> >
> > I would encourage the authors to release the dataset the research community and make sure that the data is clean and well prepared. In addition, please run simple baselines to check that there are no biases in the dataset that can be exploited by overly simple/degenerate solutions.

---

### Author Response · Authors · 2018-11-26
**Authors' comment**

Dear Editor and Referees,

Thank you very much for your insightful comments! We are happy to answer the concerns raised in the reviews. We have also prepared and uploaded an updated version of the paper, with all new text also shown below.

---

### Meta-Review · Area_Chair1 · 2018-12-14
**introducing a non conclusively useful dataset**

**Confidence:** 4
**Recommendation:** Reject

**Metareview:**

The paper introduces a new dataset that contains multiple landings from the X plane simulator, and each includes readings from multiple sensors for aircraft landing. The  paper also trains a set of self-supervised methods presented in previous works in order to learn sensory representations, and evaluates the learnt representations in terms of disentanglement and re-purposing to a discriminative task.
 Though the evaluations presented are interesting, they are not convincingly useful, as noted by the reviewers. Overall, it is not clear why this dataset is particularly well suited for representation learning. Furthermore, it is difficult to evaluate representation learning methods without relating them to an end-task, e.g., that of landing the aircraft.
The paper writing would also benefit from restructuring and improving on English expressions. In particular, the conclusion section contains half-finished sentences.